# Recent Advances in Improved Anticancer Efficacies of Camptothecin Nano-Formulations: A Systematic Review

**DOI:** 10.3390/biomedicines9050480

**Published:** 2021-04-27

**Authors:** Maryam Ghanbari-Movahed, Tea Kaceli, Arijit Mondal, Mohammad Hosein Farzaei, Anupam Bishayee

**Affiliations:** 1Medical Technology Research Center, Health Technology Institute, Kermanshah University of Medical Sciences, Kermanshah 6734667149, Iran; maryam.gh.movahed@gmail.com; 2Department of Biology, Faculty of Science, University of Guilan, Rasht 4193833697, Iran; 3Lake Erie College of Osteopathic Medicine, Bradenton, FL 34211, USA; tkaceli1@gmail.com; 4Department of Pharmaceutical Chemistry, Bengal College of Pharmaceutical Technology, Dubrajpur 731123, India; juarijitmondal@gmail.com

**Keywords:** cancer, camptothecin, natural products, nano-targeted therapy

## Abstract

Camptothecin (CPT), a natural plant alkaloid, has indicated potent antitumor activities via targeting intracellular topoisomerase I. The promise that CPT holds in therapies is restricted through factors that include lactone ring instability and water insolubility, which limits the drug oral solubility and bioavailability in blood plasma. Novel strategies involving CPT pharmacological and low doses combined with nanoparticles have indicated potent anticancer activity in vitro and in vivo. This systematic review aims to provide a comprehensive and critical evaluation of the anticancer ability of nano-CPT in various cancers as a novel and more efficient natural compound for drug development. Studies were identified through systematic searches of PubMed, Scopus, and ScienceDirect. Eligibility checks were performed based on predefined selection criteria. Eighty-two papers were included in this systematic review. There was strong evidence for the association between antitumor activity and CPT treatment. Furthermore, studies indicated that CPT nano-formulations have higher antitumor activity in comparison to free CPT, which results in enhanced efficacy for cancer treatment. The results of our study indicate that CPT nano-formulations are a potent candidate for cancer treatment and may provide further support for the clinical application of natural antitumor agents with passive targeting of tumors in the future.

## 1. Introduction

Systematic research in the tumor field has indicated that a set of multifarious processes, including quick spreading and uncontrollable multiplication of abnormal cells, results in the formation of malignant tumors with the potential for metastasis [1,2]. There are numerous diverse methods for cancer treatment, nonetheless, some might be ineffective because of the adverse side effects as well as augmented resistance to conventional antitumor drugs.

Natural compounds represent a valued resource in the discovery and development of novel drugs, mainly those used for cancer treatment [3,4,5], and they might act as a safe substitute for various synthetic drugs used in existing clinical therapies [6]. Evidence showed that alkaloids are important sources for developing plant-based antitumor drugs. Alkaloids are found generally in plants and are mainly present in blooming plants [7]. Screening for new agents has resulted in the detection of novel alkaloids that indicated potent apoptotic and antineoplastic capabilities in various cancer cells [8]. Alkaloids, such as vinblastine and camptothecin, have already been developed into antitumor drugs [9,10].

Camptothecin (CPT), a wide-spectrum antitumor agent, was primarily isolated from *Camptotheca acuminata* (family: Nyssaceae), a tree native to Tibet and China, which has been widely used in traditional Chinese medicine [11]. Various analogs of CPT are used in treating colon, ovarian, and small-cell lung cancer [7,12]. CPT’s mode of action involves suppression of the topoisomerase I enzyme (Topo I). Human Topo I is a critical enzyme involved in forming nonreversible and covalent Topo I-DNA complexes during replication of DNA and leads to strand breaks and subsequent apoptosis induction [13]. Additionally, several studies have shown that different synthetic analogs and semisynthetic derivatives of CPT act as topoisomerase inhibitors by changing ferment catalytic activity via stabilizing the covalent DNA-protein complexes [14,15,16,17,18].

The promise that CPT holds in therapy is restricted via factors that include excessive toxicity, lactone ring instability, and water insolubility, which restricts the drug oral solubility and bioavailability in blood plasma [19]. Accordingly, an efficient method for overcoming these challenges is encapsulating CPT into different nano-sized delivery vehicles [20].

Nanotechnology aims at delivering drugs more effective to their target for treating malignancies [21]. Nowadays, nanotechnology-based delivery systems have gained tremendous attention as a strategy for overcoming the challenges related to bioavailability, solubility, distribution, toxicity, and targeting of classical chemotherapeutic agents as well as antitumor natural products [22,23,24,25]. Therefore, novel approaches involving pharmacological and low doses of CPT, either alone or combined with nanoparticles, could have potent anticancer activity in vitro and in vivo. 

Even though there are a small number of publications on the overview of CPT nano-formulations in cancer, these papers are narrative reviews or reviews of the antitumor activities of nano-CPT in a limited number of cancers [26]. Henceforth, a comprehensive and critical systematic review on the antitumor capability of CPT nano-formulations within different cancers has not been conducted before. Consequently, this review aims to provide an up-to-date, comprehensive, and critical assessment of the antitumor ability of nano-CPT in various cancers as a novel and more efficient natural product-based anticancer therapeutic agent.

### 1.1. Natural Nano-Formulations and Cancer Treatment

Natural products particularly secondary metabolites have massive chemical and structural diversity along with enduring to instigate new findings in biology, medicine, and chemistry because of their therapeutic potential [27,28]. A broad variety of anticancer properties have been attributed to these plant-derived agents, including antiproliferative, antioxidant, antimetastatic, anti-inflammatory antiangiogenic, and proapoptotic activities [29,30]. Nevertheless, several other physicochemical properties, such as weak stability, low aqueous solubility, short half-life, low bioavailability, and rapid clearance, have severely restricted their uses in clinical settings [31,32].

For achieving the maximum therapeutic profits of these natural compounds, nano-sized carriers have been used for direct delivery of parent compounds to the locations where they are required, like malignant tissues to display their potential antitumor activities. Such targeted therapy could be essential for minimizing possible systemic toxicity and optimizing efficacy to increase the clinical outcomes [33]. Various nano-formulations have been used in drug delivery research, such as liposomes, solid lipid NPs, double emulsions, protein-based systems, cyclodextrins, and chitosan [24].

Alkaloids are mostly tetracyclic, tricyclic, and bicyclic derivatives of the molecule quinolizidine, mainly found in the Leguminosae family. The tumor suppression role of alkaloids nano-formulations is indicated in multiple in vitro and in vivo studies. One study used PLGA-PEG-folate (PLGA-PEG-FOL) NPs for targeted delivery of a CPT analog, 7-ethyl-10-hydroxycamptothecin (SN-38), and demonstrated significant anticancer activity in HT-29 colon tumor-bearing nude mice [34]. In another study, albumin NPs were loaded with the antitumor drug tamoxifen (TMX). The cellular uptake of NPs was observed in HeLa and MCF-7 cell lines and TMX-loaded NPs showed greater antitumor activity compared to the free drug [35]. Moreover, the cytotoxic activity of irinotecan nano-formulations against cancer has been reviewed previously [36].

Altogether, NPs derived from natural sources are considered cost-effective and more secure than synthetic NPs and offer protective and therapeutic activities with low cytotoxicity [37].

### 1.2. Camptothecin: Sources, Chemistry, and Pharmacology

CPT, an indole alkaloid with a pentacyclic ring (classified as pyrrolo [3,4-*b*]quinoline), was primarily isolated from *Camptotheca acuminata* (Nyssaceae) [1]. This alkaloid is also synthesized in other plants, including *Ervatamia heyeano* [38], *Nothapodytesfeotida* [39], and in several species of the genus Ophiorrhiza [40]. In 1966 CPT structure was determined using a combination of X-ray and Nuclear Magnetic Resonance approach (Figure 1) [41]. CPT has two remarkable chemical properties: (1) Its lack of important alkalinity makes it act as a neutral molecule, and (2) the C-20 tertiary alcohol presence imparts an uncommon electrophilicity to the lactone carbonyl group, possibly through a strong intramolecular H-bond [42].

CPT has been indicated to suppress different cancers via various mechanisms both in vitro and in vivo and rapidly entered the clinical examination (Figure 2). The most noticeable activity of CPT is the Topo I suppression which is a molecular basis of its antitumor properties [43]. The structural models show that CPT non-covalently binds to Topo I–DNA binary complex. The structure-activity relationships offer insight into a potential mechanism of Topo I suppression via CPT and its derivatives [44].

Despite the significant antineoplastic activity and an exclusive mechanism of action, CPT displays various unwanted properties which hinder its clinical application. First, the very low water solubility of CPT complicates its administration. Another challenge related to CPT is the α-hydroxy lactone ring (ring E) which opens under physiological conditions, resulting in the CPT carboxylate open form. Carboxylic acid or its sodium salt, even though soluble, has considerably lower antitumor potential as compared to CPT. Furthermore, this ionic form favorably binds to the human serum albumin, lowering the accessible drug concentration [45,46].

Even though the antitumor properties of CPT stimulated a substantial research interest, other properties of CPT including insecticidal or antiviral activities have also been investigated. Though semisynthetic CPT, such as irinotecan and topotecan, hold their position in chemotherapy, various other anticancer agents are established which are now in different phases of the preclinical or clinical development. Moreover, current research on CPT involves their novel formulations (particularly nano-formulations) for optimizing the stability, delivery, and reducing toxicity [47].

## 2. Methodology for Literature Search on Camptothecin Nano-Formulations and Cancer

The current study was conducted following the Preferred Reporting Items for Systematic Reviews and Meta-Analysis (PRISMA) guidelines [48]. The purpose of this paper is to offer a systematic review of in vivo studies for examining the impact of CPT nano-formulations on cancer. Various electronic scholarly databases, including Scopus, PubMed, ScienceDirect, were explored and related studies in the English language only were collected up to March 2021. The search syntax included “Camptothecin” AND “tumor” OR “cancer” OR “neoplasm” OR “malignancy” OR “carcinoma” AND “nano”. The primary search was performed by two researchers separately, and unrelated studies were excluded based upon their titles and abstracts. Review articles, meta-analyses, books, book chapters, conference abstracts, case reports, clinical trials, and non-English articles were also excluded. Between the initial 2586 studies that were collected through electronic search, 99 were excluded because of the duplicated results, 629 were omitted due to paper type, 905 review papers were ruled out, and 564 were considered unrelated based on abstract and/or title data. Besides, 4 were omitted as they were not in the English language. Out of 385 retrieved reports, 119 were omitted as they assessed other derivatives of CPT, 12 were excluded as they evaluated CPT rather than its nano-formulations, 41 were omitted because they assessed other biological effects of CPT rather than its antitumor impacts, 16 were omitted as they focused on other compounds, not CPT, and 120 were removed because they were in vitro studies. Finally, 82 papers were included in this systematic review as indicated in a flowchart of the selection process and literature search (Figure 3).

## 3. Anticancer Activities of Camptothecin Nano-Formulations

Nano-CPT has been indicated to suppress different cancers via various mechanisms, such as growth suppression of malignant cells, induction of apoptosis and cell cycle arrest, and modulation of oxidative stress, angiogenesis, and inflammation. The anticancer effects of CPT nano-formulations in different cancers are provided in the next sections (Table 1).

### 3.1. Bladder Cancer

Bladder cancer is the tenth most frequent cancer in women and fourth in men. The primary treatment for bladder cancer includes intravesical immunotherapy and surgery. Nevertheless, these methods show several undesirable adverse effects [130,131]. Accordingly, better therapeutic approaches are vital.

Polymeric micelles enhance the drug accumulation in cancer tissues by taking advantage of the increased permeability retention (EPR) impact [132]. In a study by Yen et al. [49], strong suppression of tumor growth without toxicity was observed in the nude mice bearing AY27 xenografts after treatment with CPT-loaded micelles. Moreover, as a wide variety of therapeutic agents could be incorporated inside polymeric micelles, designing the sensitive-reduction micelles could be readily custom-made to accomplish therapeutic necessities.

### 3.2. Brain Cancer

Gliomas are the most frequent and aggressive brain tumors, and irrespective of progress made in treatment management, they are still limited by several barriers. Therefore, different therapeutic strategies like nano-medicines are needed for treating this disease [133,134].

One study indicated that treatment of 9L tumor-bearing mice with CPT-loaded amphiphilic β-cyclodextrin NPs inhibited tumor progression and tumor volume, and increased the median survival time. This study indicated that CPT-loaded cyclodextrin NPs arise as potential delivery systems for treating cancer which is indicated to be safe, stable, and effective formulations for CPT delivery [50].

It has been shown that high-dose CPT-loaded NPs (20 mg/kg) inhibited the growth of intracranial GL261 tumors in mice, providing important survival benefits in comparison to low-dose CPT NPs or free CPT [51]. This study indicated that CPT encapsulation can increase its activity.

Another study evaluated the anticancer effect of 3 mg/kg CPD@IR780 and CPC@IR780 micelles with or without laser toward Luc-U87 tumor-bearing mice. All the treatments inhibited tumor growth, reduced side effects, and enhanced survival time. However, CPT-S-S-PEG-iRGD@IR780 micelles with laser exhibited superior antitumor activity. Thus, these results indicated that the targeting prodrug system could not only efficiently cross different barriers to reach glioma site but also considerably increase the anticancer impact with laser [52].

Studies showed that treatment of C6 tumor-bearing rats with different CPT-loaded micelles inhibited tumor progression and growth, reduced side effects, and increased median survival rate and therapeutic efficacy [53,54]. Moreover, CPT-TEG-ALA treatment inhibited tumor growth and enhanced the survival rates and therapeutic efficacy in U-87 MG tumor-bearing nude mice. This nano-prodrug method is a versatile approach to develop therapeutic NPs enabling tumor-specific treatment [55]. The non-toxic, tumor-specific targeting properties of the nano-prodrug system make it a versatile, low-cost, and safe nano-carrier for diagnostic agents, imaging agents, and pharmaceuticals.

### 3.3. Breast Cancer

Breast cancer is the utmost frequently diagnosed malignancy and is the second primary cause of mortality globally. Despite these improvements, breast cancer mortality and morbidity is very high [135]. Therefore, the application of novel methods like nano-therapies is essential for the prevention and treatment of this cancer [136].

One study indicated that in 4T1 tumor-bearing mice, the albumin/CPT-ss-EB nano-complex exhibited efficient tumor accumulation, which subsequently contributed to outstanding therapeutic efficiency [56]. In a study by Zhang [137], the MOF-2 + CPT therapeutic impact was evaluated. The result of this study demonstrated that both CPT and MOF-2 can suppress tumor growth in the mouse breast cancer model. After the combination of CPT with MOF-2, the therapeutic efficiency was noticeably improved.

Several studies showed that treatment of 4T1 tumor-bearing mice with different CPT-loaded micelles (FP-CPT, DCPT, HA-ss-CPT, DCM, and DEM micelles) inhibited the tumor growth and tumor progression and resulted in enhancement of animal survival rate and therapeutic efficacy [57,58,59,60]. PM_CPT_-co-electrospun fibers showed anticancer activity toward 4T1 tumor-bearing mice and led to the suppression of the tumor growth without toxicity [61].

Studies indicated that treating 4T1 tumor-bearing mice with various CPT-loaded NPs led to the suppression of tumor growth and reduction in tumor metastases and tumor recurrence [62,63,64]. Moreover, it has been shown that treatment with AmpF-CPT-IR820 (ACI), and PF + CPT + IR820 (PCI) inhibited tumor growth and volume in 4T1 breast tumor-bearing mice [65].

Treating 4T1 tumor-bearing mice with chitosan stabilized CPT nano-emulsions (CHI-CPT-NEs) led to the reduction of tumor growth and volume and enhancement of therapeutic efficacy [66]. Additionally, it has been shown that treatment with 1 mg/kg MDNCs suppressed tumor progression and tumor growth in 4T1 tumor-bearing mice and exhibited superior anticancer impact in comparison to individual drug treatment groups and free multi-drugs treatment groups [67].

In another study, the in vivo synergistic antitumor efficiency of HSD NGs was assessed in 4T1 tumor-bearing mice, and results indicated that HSD NGs enhanced apoptosis and tumor accumulation, and suppressed tumor growth and volume [68].

Treating MCF-7 tumor-bearing mice with various CPT-loaded micelles led to the strong inhibition of the tumor growth without toxicity [69,70,71]. Moreover, treatment with 10 mg/kg CPT@Ru-CD, VK3-CPT@Ru-CD, and VK3-CPT@RuxCD led to the tumor elimination, increased ROS levels and reduced side effects in MCF-7 tumor-bearing nude mice [72].

Several studies showed that treatment with different CPT-loaded nano-formulations led to tumor elimination and inhibited tumor growth and volume in MCF-7 tumor-bearing mice [73,74,75,76]. Moreover, CPT-loaded MrGO-AA-*g*-4-HC increased the synergistic antitumor efficacy and apoptosis in rats bearing MCF-7 tumor cells [77]. Another study showed that treating MCF-7 tumor-bearing mice with GNS-CB[7]-CPT with or without laser resulted in the elimination of the tumor. It has been shown that GNS-CB[7]-CPT with laser irradiation exhibited severe necrosis and had a greater antitumor activity [78].

Studies showed that an increase in survival rate and tumor inhibition has been observed in MDA-MB-231 tumor-bearing mice after treatment with CPT-HGC NPs [79] and CPT-P-HA-NPs [80]. Moreover, treating MDA-MB-231BO tumor-bearing mice with 7.5 mg/kgpSiNP + CPT + Ab and pSiNP + CPT led to the reduction in tumor size and inhibition of metastatic spread [81]. Another study showed that treating MDA-MB-231 tumor-bearing mice after treatment with CPT-pH-PMs, led to strong suppression of the tumor growth and reduction in side effects [82].

Treatment of EMT6 tumor-bearing mice with 1 mg/kg CPT/DOX-CCM and CPT/DOX-NCM, for 24 days led to tumor inhibition and reduced tumor size and volume [83]. CPT-loaded NPs (^CPT^NV-P_mic_) exhibited considerable anticancer activities against the Ehrlich ascites carcinoma (EAC) mice model and reduced tumor necrosis and tumor growth [84]. In another study, the anticancer efficacy of CPT excipient formulation and CPT encapsulated in E_3_ PA nanofibers was assessed in BT-474 tumor-bearing athymic nude mice, and results demonstrated that these nanofibers inhibited tumor growth and tumor progression [85]. Such nano-structures provided CPT protection from the external environment and increased CPT anticancer activity.

### 3.4. Cervical Cancer

Cervical cancer is the second highest cause of mortality amongst women [138]. Even though chemotherapy is the primary method for the treatment of cervical cancer, new approaches are essential for the enhancement of the efficacy of existing cervical cancer therapy [139].

Studies showed that treatment with different CPT-loaded NPs (CPT@IrS***_x_***-PEG-FA NPs andPDA@PCPT NPs) led to the suppression of tumor progression and tumor growth and increased the therapeutic efficiency in HeLa tumor-bearing mice [86,87]. Moreover, another study showed that treating Hela tumor-bearing mice with 4 mg/kg CPT-PCB-based lipoplexes inhibited tumor progression and tumor growth, and displayed a synergistic tumor inhibition [88].

In a study by Jiang et al. [89], the synergistic effect of RLS/siPLK1+ CPT on tumor growth suppression was evaluated in HeLa tumor-bearing nude mice. Results of this study indicated that RLS/siPLK1+ CPT has a greater anticancer activity compared to either agent alone. These data indicate that R_2_SC/siPLK1 can efficiently inhibit tumor growth via silencing the siPLK1 gene and controlling the drug release and through inducing tumor cells apoptosis, contributing to a synergistic impact of siPLK1 and CPT.

In another study, the antitumor efficiency of CPT-DNS-DCM was examined on HeLa tumor-bearing nude mice. Results indicated that the prodrug considerably inhibited tumor growth and tumor progression [91]. Moreover, treating HeLa tumor-bearing nude mice with EuGd-SS-CPT-FA-MSNs led to the destruction of tumors and reduction of tumor growth and volume. EuGd-SS-CPT-FA-MSNs might offer a beneficial theranostic nano-platform for inhibiting tumor growth in vivo [90]. The results of in vivo studies indicate that the functionalized MSNs might be effectively used as a platform for targeted therapy.

### 3.5. Colon Cancer

Colorectal cancer (CRC) is one of the utmost common malignant tumors. The main approaches for the treatment of CRC are surgery, chemotherapy, and radiotherapy [140]. Yet, due to the difficulties resulting from drug resistance, the application of multifunctional nano-medicines could be a viable therapeutic approach [140].

One study showed that transformative CPT-ss-EB nanomedicine considerably suppressed tumor progression and drastically decreased side effects in HCT116 tumor-bearing mice [56]. Another study showed that treatment of SW620 tumor-bearing nude mice with 10 mg/kg CPT-loaded CMD NPs for 30 days resulted in the suppression of tumor growth and progression. CPT-loaded CMD NPs showed considerably higher antitumor activity than free CPT and empty CMD NPs in mouse xenograft models, indicating the synergistic therapeutic impacts of CPT with CMD [92].

In one study the antitumor activity of CPT&Ce6 and HBPTK-Ce6@CPT with or without laser irradiation in HT29 tumor-bearing nude mice has been evaluated. All the treatments inhibited tumor growth and volume. However, HBPTK-Ce6@CPT with 660 nm laser irradiation had greater anticancer activity in comparison to other groups [93].

Studies showed that treatment of HCT116 tumor-bearing mice with different CPT-loaded NPs resulted in the elimination of tumor, reduction of tumor growth and volume, tumor recurrence, and increase of therapeutic efficacy and apoptosis [94,95,96,97]. Moreover, HRC@F127 NPs indicated synergistic therapeutic efficiency and effectual tumor accumulation with minimal side effects and longer survival in mice [98].

Treatment of HT-29.Fluc tumor-bearing mice with 0.8 mg/kg SNP-CPT-Cy5.5 and SNP-CPT NPs led to the tumor elimination and inhibited tumor growth and progression, while considerably decreasing the systemic toxicity associated with CPT administration. These results demonstrate that the SNP-CPT NPs can be used as a potent drug delivery system for CPT-based anticancer treatments [99].

In a study by Yao et al. [100], the in vivo anticancer activity of nano-CPT compared with that of topotecan was assessed against HCT-8 tumor-bearing mice. Results indicated that nano-CPT had the same in vivo anticancer activity with TPT and lower toxicity. The study indicated that nano-CPT is a new promising formulation with high anticancer efficiency and low toxicity [100].

Gd(DTPA-CPT) NPs showed significant cytotoxicity against LoVo tumor-bearing nude mice and led to the tumor growth suppression with reduced adverse effects and insignificant chronic toxicity [101]. Another study showed that treatment with 3 mg/kg CPT@Dod-ND-SPs led to suppression of tumor growth and tumor volume in HT-29 tumor-bearing mice. The results indicated that ND-SPs might serve as a nanomedicine with major therapeutic potential [102].

Several studies indicated that treating C26-tumor-bearing mice with different CPT nano-formulations resulted in the elimination of tumor growth and volume, enhancement of therapeutic efficacy, and reduction in tumor progression [103,104,105,106]. Furthermore, it has been shown that iRGD-PEG-NPs considerably increased the therapeutic efficiency of CPT via inducing tumor cell apoptosis in comparison to PEG-NPs [107]. These results indicate a promising approach for small molecular nano-drug delivery systems with noticeable antitumor efficacy for clinical applications.

### 3.6. Liver Cancer

Hepatocellular carcinoma (HCC) is the second leading cause of death in the world. Irrespective of the advancement in existing HCC treatments, there has been a continuous increment in the incidence rate of this cancer [141].

In a study by Wen et al. [108], the in vivo synergistic anticancer efficacy of MGO@CD-CA-HA/CPT with or without NIR was assessed in BEL-7402 tumor-bearing nude mice. Results showed that MGO@CD-CA-HA/CPT + NIR had a stronger inhibitory effect than MGO@CD-CA-HA/CPT, exhibiting the meliority of CPT combination therapy. Thus, this study presented a potent multiple-targeted nanocarrier for liver cancer chemo-photothermal combination therapy.

Moreover, studies showed that treatment with different CPT prodrugs (CCLM and RGD-prodrug) led to tumor elimination and suppressing tumor progression and growth in HepG2 and H22 tumor-bearing mice [109,110]. These prodrug nano-platforms showed considerable in vivo anticancer efficacy, without displaying considerable systemic toxicity.

In one study the antitumor activity of IR780-LA/CPT-ss-CPT NPs were investigated towards Hep1–6 tumor-bearing mice. It was noticed that the tumor volume and growth in all groups were suppressed. Nevertheless, the tumor growth in the IR780-LA/CPT-ss-CPT NPs + laser group was the most strictly suppressed. Consequently, the IR780-LA/CPT-ss-CPT NPs were indicated to be a brilliant fluorescence imaging-guided, redox-responsive, and increased synergistic chemo-photothermal therapy nanoplatform toward tumors [111].

Treatment of HepG2 tumor-bearing mice with P (CPT-MAA) nanogels without SeS and P(CPT-MAA) prodrug nanogels exhibited superior antitumor activity without observed side effects. However, P(CPT-MAA) prodrug nanogels indicated the greatest efficacy in tumor growth suppression. Henceforth, the P(CPT-MAA) nanogels might be a potent delivery system for anticancer agents [112].

JP@EF and JNM@EF showed cytotoxic effects against H22 tumor-bearing mice, while the most considerable suppression of tumor growth was identified for JNM@EF treatment. It was demonstrated that the JNMs self-propelled tissue distribution and gradual release increased the tumor growth suppression [113].

In another study, the anticancer effects of UCNP@mSiO2–NBCCPT @(DHMA)/β-CD-PEG, UCNP@mSiO2-NBCCPT@(DHMA)/β-CD-PEG-LA, and UCNP@mSiO2-NBCCPT/β-CD-PEG were assessed in HepG2 tumor-bearing mice. All the treatments inhibited tumor growth, while UCNP@mSiO2-NBCCPT/β-CD-PEG-LA@DHMA had greater antitumor activity and ameliorated side effects. Therefore, it showed more beneficial for prolonging the survival of tumor-bearing mice [114]. Overall, the results indicated that the nano-drug delivery systems have the ability to overwhelming the CPT pharmacokinetic limitations, highlighting its anticancer effects.

### 3.7. Lung Cancer

Lung cancer is one of the leading causes of tumor-related mortality in the world [142]. The high lung cancer death rates are probably because of difficulties related to a high metastatic potential and diagnosis [143]. Accordingly, developing non-toxic alternative treatments for improving lung cancer responsiveness to chemotherapy is essential.

Even though CPT is a renowned antitumor agent, it typically needs high and multiple doses for achieving a satisfactory therapeutic impact. One study investigated the antitumor impact of CPT-loaded PPBS NPs in LLC tumor-bearing mice. Results of this study indicated that CPT-loaded PPBS NPs reduced the tumor volume and had a longer circulation time, and substantially improved anticancer efficiency in vivo compared to free CPT [115].

NCssG NPs and CPT-ss-GEM nanowires (CssG NWs) showed cytotoxic effects against A549 tumor-bearing mice and suppressed tumor growth and volume. Results also showed that with the same amount of CPT-ss-GEM prodrug, NCssG NPs showed much higher in vivo tumor inhibition efficiency than CssG NWs. Due to the superiority of this combo-nanomedicine, such as EPR effect, synchronous dual drug action, prolonged blood circulation, very potent antitumor efficiency was achieved both in vitro and in vivo [116].

Studies showed that treatment of A549 tumor-bearing nude mice with different nano-formulations of CPT resulted in tumor elimination, reduction in tumor growth and volume, and enhancement of therapeutic efficacy [117,118]. These combination approaches hold great promise for the future potential application in cancer therapy.

Another study indicated that treatment with ZTC-NMs delayed tumor growth and prolonged the survival time in A549 tumor-bearing mice. Also these results showed that ZTC-NMs had a greater antitumor activity, which can be ascribed to the tumor-targeted drug delivery [119].

In one study the antitumor effect of Ce6-CPT-UCNPs with or without laser irradiation was evaluated in NCI-H460 tumor-bearing nude mice. Results showed that treatment with Ce6-CPT-UCNPs + laser irradiation inhibited tumor growth, tumor recurrence, and metastasis. In contrast, treatment with Ce6-CPT-UCNPs and no laser irradiation did not show antitumor activity, because CPT and Ce6 in the Ce6-CPT-UCNPs cannot be released out, and cannot eliminate cancer cells [120].

Treatment of NCI-H460 tumor-bearing nude mice with a new mitochondria-targeting drug delivery system, ZnPc/CPT-TPPNPs or ZnPc/CPT-NH2NPs led to the suppression of tumor growth, tumor recurrence, and metastasis and did not result in considerable side effects in vivo. These results demonstrated that surface modification of the NPs with triphenylphosphine cations simplified effective subcellular delivery of the photosensitizer to mitochondria [121].

In another study, the in vivo anticancer efficiency of 2OA-CPT/NAs and OA-CPT/NAs was compared using the LLC tumor-bearing mice. It is found that treatments with different CPT formulations led to a considerably delayed tumor progression. However, in comparison with OA-CPT/NAs, 2OA-CPT/NA exhibited a greater potent anticancer activity. These results revealed the critical role of DHP hydrophobicity in impacting the DHP NA in vivo anticancer efficacy [122].

### 3.8. Ovarian Cancer

Ovarian cancer represents a group of neoplasms, and it is one of the utmost lethal female reproductive system tumors [144]. The conventional approaches for the treatment of this cancer are platinum-based chemotherapy and surgical cytoreduction, which have their challenges [145]. Consequently, investigating new agents with increased efficiency and reduced toxicity can open up new pathways for ovarian cancer treatment.

In one study CPT was encapsulated into plain NOBs (NOB–CPT) and ZH2-displayed NOBs (ZH–NOB–CPT) and then administered to SKOV3 tumor-bearing nude mice. Results showed that treatment with 0.5 mg/kg ZH–NOB–CPT significantly inhibited tumor growth and volume. In contrast, NOB–CPT-treated mice did not show any antitumor effects. This study indicates that ZH2-tagged NOBs selectively deliver CPT into the human epidermal growth factor receptor 2/neu-positive cancerous site. Taken together, the result shows the NOB’s potential for targeted delivery of hydrophobic drugs [123].

### 3.9. Pancreatic Cancer

Pancreatic cancer is a disease with a high mortality rate and a very poor prognosis. Various biological and physical barriers make this tumor very hard to treat with conventional chemotherapy. Hence, more effective strategies, such as nano-therapies, are crucial to fulfill the immediate necessity for more effective pancreatic cancer treatment [146,147].

One study evaluated the therapeutic effects of EGGPTCPT, PEGPTCPT, and GSHPTCPT in BxPC-3 tumor-bearing mice. All the treatments resulted in tumor inhibition, while GSHPTCPT displayed considerably greater anticancer activity than PEGPTCPT and EGGPTCPT. There was no considerable difference between PEGPTCPT and EGGPTCPT in terms of tumor weight and volume. This study indicates the high efficacy of an active tumor penetrating dendrimer-drug conjugate for PDA therapy [124].

Another study showed that treatment with 2 mg/kg CPT-loaded αDR5-NPs and CPT-loaded nude NPs noticeably suppressed tumor growth rates and induced tumor regressions in MIA PaCa-2 and PANC-1 xenografts in mice. The effects were more considerable in the MIA PaCa-2 cells where delivery of CPT-loaded αDR5-NPs resulted in considerable tumor regressions, whereas the CPT-loaded nude NPs only resulted in growth retardation. These results indicate the CPT-loaded αDR5-NPs potential for pancreatic cancer treatment [125].

In a study by Wang et al. [126], the DPPSC micelles in vivo anticancer efficiency were investigated in the pancreas cancer treatment. Results of this study showed that treatment of PANC-1 tumor-bearing nude mice with DPPSC + SL and DPPSC + LL micelles remarkably inhibited the tumor growth and tumor volume. Moreover, it has been demonstrated that the mice treated with the DPPSC + SL group led to a more significant reduction in the tumor growth, compared to the DPPSC + LL group. Hence, this system indicated a high anticancer impact in PANC-1 tumor-bearing mice because of the combination of PDT and enhanced chemotherapy. This study presents a novel approach for the design of the stepwise multiple stimuli-responsive nano-carrier with the variation of biological signals to maximize the treatment outcomes while minimizing the side effects of therapeutic agents.

### 3.10. Prostate Cancer

Prostate cancer is one of the most common cancers in men [148]. The combined consumption of nutraceutical agents and anticancer drugs is an excellent strategy to enhance the therapeutic antitumor effects as well as the facilitation of side effects of chemotherapy and drug resistance [149].

One study indicated that treatment of U14 tumor-bearing nude mice with CPT-HA and CPT-HA@IR825 with or without laser led to tumor elimination. In vivo results demonstrate that after treating tumors with CPT-HA or CPT-HA@IR825 without laser, the tumor volume was only partially inhibited. In sharp contrast, the tumor growth from the CPT-HA@IR825-treated mice upon laser irradiation has been effectively suppressed, indicating that the multifunctional polymeric prodrug NPs are very effective for cancer treatment, while having reduced cytotoxicity to the healthy organs [127].

Another study indicated that treatment with P(OEGMA-co-CPT-co-G3-C12) or P(OEGMA-co-CPT) for 32 days inhibited tumor volume and tumor growth in DU145 tumor-bearing BALB/c mice. Results of this study indicated that P(OEGMA-co-CPT-co-G3-C12) had a greater antitumor activity compared to P(OEGMA-co-CPT). Furthermore, P(OEGMA-co-CPT-co-G3-C12) indicated minimal toxicity and improved antitumor activity compared to free CPT. Henceforth, P(OEGMA-co-CPT-co-G3-C12) can be a potent drug in androgen-independent prostate cancer treatment [128].

### 3.11. Skin Cancer

Melanoma is the primary cause of death from skin cancer [150]. Generally, treating melanoma with chemotherapy provides very limited beneficial effects in overall survival and very low response rates. Consequently, multiple targeted therapeutic strategies have been assessed [151].

In a study by Hu et al. [129], nude mice bearing B16 tumor cells were treated with PEG-SeSe-CPT/CUR and PEG-SeSe-CPT for 21 days. In vivo assay showed that both of the treatments could inhibit B16 tumor growth and volume. This result also showed that the PEG-SeSe-CPT/CUR co-delivery system had a much better inhibition effect than either agent alone, suggesting a synergistic therapeutic effect.

## 4. Pharmacokinetics and Toxicity of Nano-Camptothecin

The plasma drug time-concentration profile of CPT was constructed after the oral administration of 5 mg/kg pure CPT suspension and CPT encapsulated poly (methacylic acid-co-methyl methacrylate) nano-formulation to rats. The results demonstrated a considerable difference between the CPT nano-formulation and free CPT pharmacokinetic profiles. After oral administration of free CPT, the drug was detected rapidly in plasma in the initial hours, ascribed to the greater CPT permeability coefficient in the upper GIT. Subsequently, the drug-plasma concentration reduced rapidly to undetectable levels after 8 h. In the case of CPT nano-formulation, the maximum CPT level was reached at 12 h after oral administration as the polymer present in the nano-formulation enhanced the CPT residence time and then gradually reduced over the next 12 h, which demonstrated the released drug prolonged residence time in the colon with drug slow leaching to systemic circulation because of low permeability and compromised surface area [152,153].

After a single oral administration (1.5 mg/kg) of CPT and CPT nanocrystals in Sprague-Dawley rats, it was reported that in comparison to CPT salt solution, CPT nanocrystal’s area under the curve (AUC) value was lower, but the distribution half-life (t1/2α) was higher, showing prolonged circulation via the nanocrystals. It seems to be plausible that due to the poor solubility, CPT nanocrystals gradually dissolved and released free drug molecules [154]. Therefore, concerning extending blood circulation, the CPT nanocrystals might provide considerable benefits [155].

For evaluating the subacute toxicity of CPT NPs, ICR mice were administered with 0.4 mL/20 g. body weight NPs through i.p. injection. There was no treatment-related death in the CPT NPs treated mice at any of the doses tested. Nevertheless, the free CPT group body weight was considerably lower than information from the control group and the two other CPT groups [156].

In an inorganic–organic model, CPT was primarily incorporated into micelles derived from negatively charged biocompatible surfactants, such as sodium cholate, and these negatively charged micelles were next encapsulated in NPs of magnesium–aluminum layered double hydroxides (LDHs) via an ion exchange process. Although these nano-complexes exhibited lower cytotoxicity than CPT alone, they act as a potent biocompatible model for delivering CPT, enabling the administration of drugs in a dose-controlled manner [157].

After CPT-loaded GNR@SiO2−tLyP-1 treatment, the systemic toxicity of CPT on human mesenchymal stem cells was reduced, since the GNR@SiO2−tLyP-1 selectively penetrated the cancer cells, and CPT was released into the blood or culture medium [158].

## 5. Conclusions and Future Directions

Natural products with remarkable chemical diversity have been examined in the treatment of human malignancies and particularly in cancer treatment. CPT, a natural plant alkaloid, has indicated strong antitumor activities via targeting intracellular Topo I. CPT alone or in combination with other anticancer drugs might be beneficial for cancer treatment. Studies indicated that CPT nano-formulations have higher antitumor activities in comparison to free CPT and lead to better bioefficacy for the treatment of cancer. CPT can affect different cancer types, including bladder, brain, breast, cervical, colon, leukemia, liver, lung, melanoma, and prostate cancer. It has been shown that CPT employs multiple mechanisms for suppressing cancer initiation, progression, and promotion through modulating different dysregulated signaling cascades implicated in proliferation, invasion, inflammation, cell survival, metastasis, and apoptosis (Figure 4).

The promise that CPT holds as an antitumor agent in therapy is restricted by factors that include lactone ring instability and water insolubility, which restricts the drug oral solubility and bioavailability in blood plasma. Novel approaches including pharmacological and low doses of CPT in combination with NPs have indicated potent anticancer potential in vivo. The results of our study indicate that CPT nano-formulations is a potential candidate for cancer treatment, and might provide further support for developing nano-CPT as a promising agent for cancer treatment.

## Figures and Tables

**Figure 1 biomedicines-09-00480-f001:**
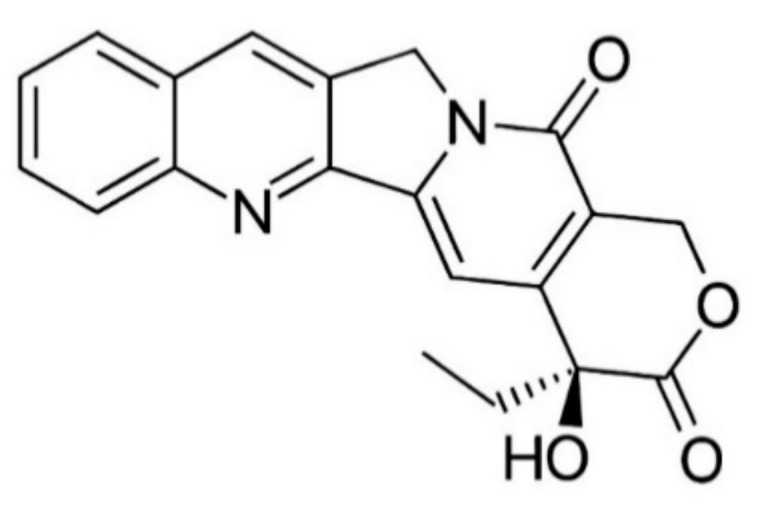
The chemical structure of CPT (C_20_H_16_N_2_O_4_, molecular weight: 348.4 g/mol).

**Figure 2 biomedicines-09-00480-f002:**
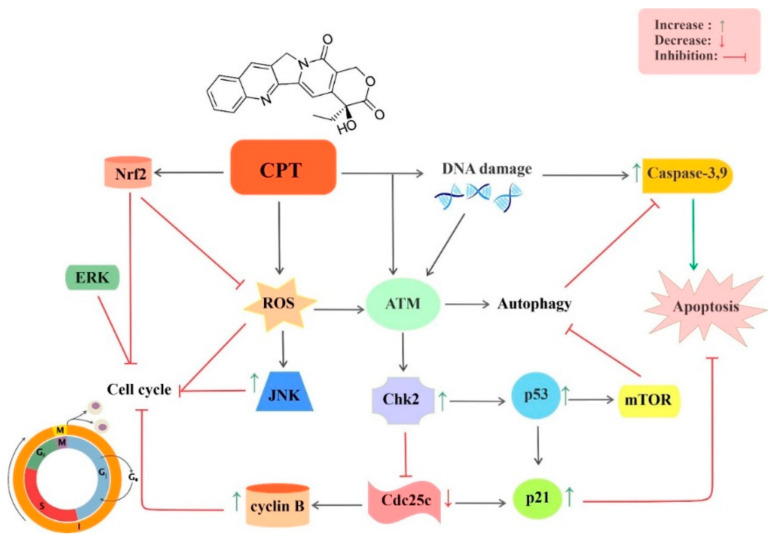
Molecular mechanisms underlying the anticancer effect of CPT. Abbreviations: ATM, ataxia telangiectasia mutated gene; Cdc25c, cell division cycle 25C; Chk2, checkpoint kinase 2; CPT, camptothecin; ERK, extracellular signal-regulated kinase; JNK, c-Jun N-terminal kinase; mTOR, mammalian target of rapamycin; Nfr2, nuclear factor erythroid 2–related factor 2; p21, tumor protein p53; p53, tumor protein p53; ROS, reactive oxygen species.

**Figure 3 biomedicines-09-00480-f003:**
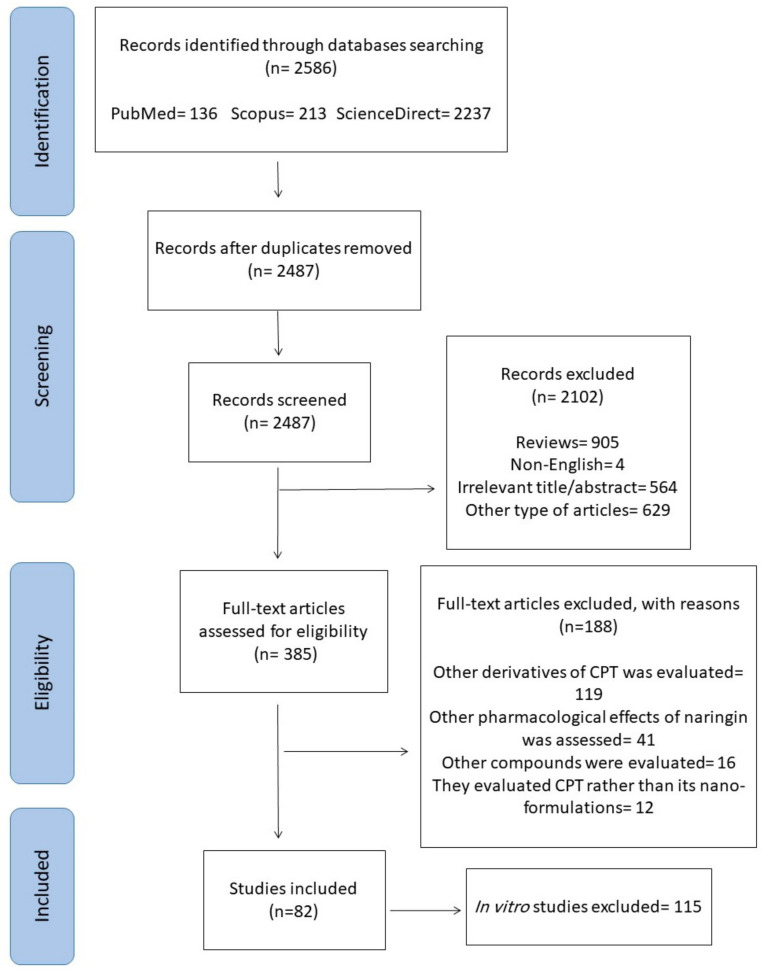
The PRISMA flow chart of the selection process for the included studies.

**Figure 4 biomedicines-09-00480-f004:**
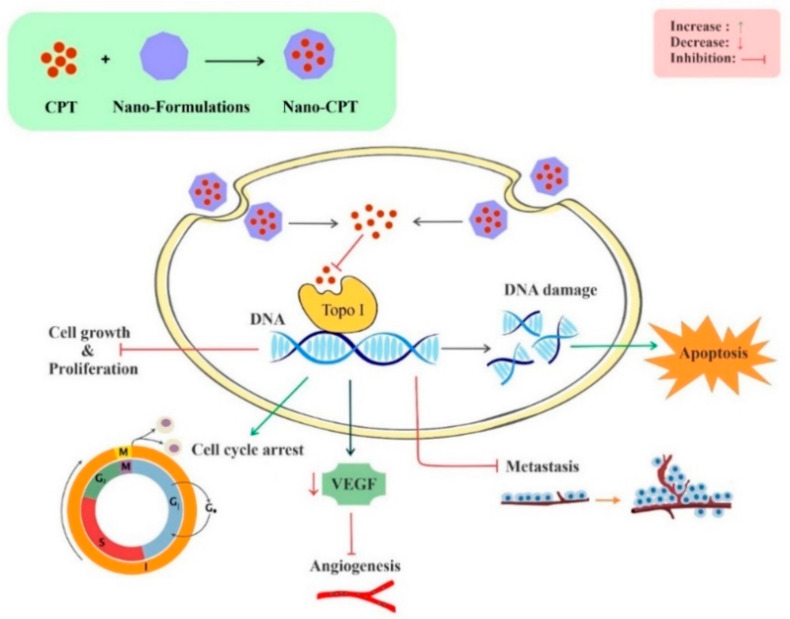
Molecularmechanisms underlying antitumor effects of CPT nano-formulations. Abbreviations: CPT, camptothecin; Topo I, topoisomerase I; VEGF, vascular endothelial growth factor.

**Table 1 biomedicines-09-00480-t001:** Potential antitumor impacts of CPT nano-formulations based on in vivo studies.

Cancer Type	Animal Model	Nano-Formulation	Dose & Duration	Anticancer Effects	References
Bladder	Nude mice bearing subcutaneous AY27 xenografts	CPT-loaded micelles	5–10 mg/mL for 16 days	↑Tumor elimination, ↓tumor growth	[49]
Brain	9 L tumor-bearing mice	CPT-loaded amphiphilic β-cyclodextrin NPs	40 days	↓Tumor progression, ↓tumor growth, ↓tumor volume, ↑median survival time	[50]
Brain	GL261-luc2 tumor-bearing C57BL/6 albino mice	CPT-loaded PLGA NPs	10–20 mg/kg for 60 days	↑Tumor elimination, ↓tumor progression, ↓tumor growth	[51]
Brain	Mice bearing intracranial Luc-U87 glioma	CPC@IR780 micelles and CPD@IR780 micelles	3 mg/kg for 28 days	↑Survival time, ↓tumor growth, ↓side effects	[52]
Brain	C6 tumor-bearing rats	CPT-loaded Bom/PEG-PCL-Tat micelles	1.2 mg/kg for 60 days	↑Therapeutic efficacy, ↑tumor elimination, ↓tumor progression, ↓tumor growth	[53]
Brain	Rats bearing intracranial C6 glioma tumors	CPT-loaded MPEG-PCL and CPT-loaded MPEG-PCL-Tat micelles	1.2 mg/kg for 60 days	↑Median survival, ↓tumor growth, ↓side effects	[54]
Brain	U-87 MG tumor-bearing nude mice	CPT-TEG-ALA	4–16 mg/kg for 5 days	↑Survival time, ↓tumor progression, ↓tumor growth, ↑therapeutic efficacy	[55]
Breast	4T1 tumor-bearing BALB/c mice	CPT-ss-EB or CPT-cc-EB	3 mg/kg for 2 weeks	↓Tumor progression, ↓tumor growth, ↓side effects	[56]
Breast	4T1 tumor-bearing mice	FP-CPT/TNF micelleplexes, FP-CPT micelles	3.7 mg/kg for 25 days	↑ Animal survival, ↓tumor growth, ↓tumor volume	[57]
Breast	4T1 tumor-bearing BALB/c mice	DCPT micelles	5 mg/kg for 11 days	↓Tumor progression, ↓tumor growth, ↑therapeutic efficacy	[58]
Breast	4T1 tumor-bearing mice	HA-ss-CPT micelles	4 mg/kg for 60 days	↑Tumor elimination, ↓tumor progression, ↓tumor growth	[59]
Breast	4T1 tumor-bearing mice	DCM and DEM micelles	4 mg/kg for 60 days	↑Tumor elimination, ↓tumor progression, ↓tumor growth	[60]
Breast	4T1 tumor-bearing Balb/c mice	PM_CPT_-co-electrospun fibers	4.0 mg/kg for 30 days	↑Therapeutic efficacy, ↑tumor elimination, ↓tumor progression, ↓tumor growth	[61]
Breast	4T1 tumor-bearing mice	NP/CPT	5 mg/kg for 16 days	↑Tumor elimination, ↓tumor progression, ↓tumor growth	[62]
Breast	4 T1 tumor-bearing mice	CPT-SS-APBA, BBSA/CPT-SS-APBA NPs	5 mg/kg for 18 days	↑Tumor elimination, ↓tumor progression, ↓tumor growth	[63]
Breast	4T1 tumor-bearing mic	P@CH NPs	20–65 days	↓Tumor growth, ↓tumor metastases, ↓tumor recurrence	[64]
Breast	4T1 breast tumor-bearing mice	ACI, and PCI	0.9 mg/kg for 25 days	↑Tumor elimination, ↓tumor growth, ↓tumor volume	[65]
Breast	4T1 tumor-bearing BALB/c mice	CHI-CPT-NEs	2.5 mg/kg for 4 weeks	↑Therapeutic activity, ↑tumor elimination, ↓tumor growth	[66]
Breast	4T1 tumor-bearing mice	MDNCs	1 mg/kg for 2 weeks	↑Tumor elimination, ↓tumor progression, ↓tumor growth	[67]
Breast	4T1 tumor-bearing mice	HSD NGs	5 mg/kg for 12 days	↑Tumor accumulation, ↑apoptosis, ↓tumor progression, ↓tumor growth	[68]
Breast	MCF-7 tumor-bearing nude mice	CCO micelles	5 mg/kg for 1–14 days	↑Tumor elimination, ↓tumor progression, ↓tumor growth	[69]
Breast	MCF-7 tumor-bearing mice	CP micelles	5 mg/kg for 12 days	↑Inhibition rate, ↓tumor progression, ↓tumor growth	[70]
Breast	BALB/c nude mice bearing MCF-7 xenograft tumors	Cy@CPT or Ce@CPT micelles	1g/kg for 1–10 days	↓Tumor progression, ↓tumor growth	[71]
Breast	MCF-7 tumor-bearing nude mice	CPT@Ru-CD, VK3-CPT@RuxCD andVK3-CPT@Ru-CD	10 mg/kg for 16 days	↑ROS, ↑tumor elimination, ↓tumor progression, ↓tumor growth, ↓side effects	[72]
Breast	MCF-7 tumor-bearing mice	CPT-PGA encapsulated SNPs	13.6 mg/kg for 10 days	↑Tumor elimination, ↓tumor progression, ↓tumor growth	[73]
Breast	MCF-7 tumor-bearing nude mice	CPT/NS	5 mg/kg for 16 days	↑Tumor elimination, ↓tumor growth, ↓tumor volume	[74]
Breast	MCF-7 tumor-bearing mice	CSP-CPT	10 mg/kg for 25 days	↑Therapeutic activity, ↑tumor elimination, ↓tumor progression, ↓tumor growth	[75]
Breast	MCF-7 tumor-bearing nude mice	CCP UMs	1 mg/mL for 15 days	↑Tumor elimination, ↓tumor progression, ↓tumor growth	[76]
Breast	MCF-7 tumor-bearing rat	CPT-loaded MrGO-AA-*g*-4-HC	5 µg/kg for 6 weeks	↑Synergistic anti-tumor efficiency, ↑apoptosis, ↓tumor growth	[77]
Breast	MCF-7 tumor-bearing mouse	GNS-CB[7]-CPT	300 μg for 1–15 days	↑Tumor elimination, ↓tumor progression, ↓tumor growth	[78]
Breast	MDA-MB231 tumor-bearing nude mice	CPT-HGC NPs	10–30 mg/kg for 35 days	↑Survival rate, ↑tumor elimination, ↓tumor progression, ↓tumor growth	[79]
Breast	MDA-MB-231 tumor-bearing mice	CPT-P-HA-NPs	10 mg/kg for 40 days	↑Survival rate, ↓tumor progression, ↓tumor growth	[80]
Breast	MDA-MB-231BO tumor-bearing mice	pSiNP + CPT, pSiNP + CPT + Ab	7.5 mg/kg for 15 weeks	↓Tumor growth, ↓metastatic spread, ↓tumor size, ↑tumor elimination	[81]
Breast	MDA-MB231 tumor-bearing mice	CPT-pH-PMs	5–10 mg/kg for 50 days	↓Tumor progression, ↓tumor growth, ↓side effects, ↑survival rate	[82]
Breast	EMT6 tumor bearing Balb/c mice	CPT/DOX-NCM, and CPT/ DOX-CCM	1 mg/kg for 24 days	↑Tumor elimination, ↓tumor progression, ↓tumor growth	[83]
Breast	Ehrlich Ascites Carcinoma (EAC) mice models	CPT-loaded NPs (^CPT^NV-P_mic_)	2–3 mg/kg for 30 days	↓Necrosis, ↑tumor elimination, ↓tumor progression, ↓tumor growth	[84]
Breast	BT-474 tumor-bearing athymic nude mice	CPT encapsulated in E_3_ PA nanofibers	1.5 mg/kg for 40 days	↓Tumor growth, ↑anti-tumor activity	[85]
Cervix	Mice bearing HeLa tumors	CPT@IrS***_x_***-PEG-FA NPs	2 mg/mL for 2 weeks	↑Tumor elimination, ↓tumor progression, ↓tumor growth	[86]
Cervix	HeLa tumor-bearing mice	PDA@PCPT NPs	9.81 mg/kg for 2 weeks	↑Therapeutic efficacy, ↑tumor elimination, ↓tumor progression, ↓tumor growth	[87]
Cervix	Hela tumor-bearing mice	CPT-PCBn based lipoplexes	4 mg/kg for 28 days	↑Tumor suppression, ↓tumor progression, ↓tumor growth	[88]
Cervix	HeLa tumor-bearing nude mice	RLS/siPLK1+ CPT	14–26 days	↓Tumor growth, ↓tumor volumes, ↓side effects, ↑apoptosis	[89]
Cervix	HeLa tumor-bearing nude mice	EuGd-SS-CPT-FA-MSNs	23 days	↑Tumor elimination, ↓tumor progression, ↓tumor growth	[90]
Cervix	HeLa tumor-bearing BALB/c-nu nude mice	CPT-DNS-DCM	20 mg/kg for 12 days	↑Tumor elimination, ↓tumor progression, ↓tumor growth	[91]
Colon	HCT116 tumor-bearing mice	CPT-ss-EB or CPT-cc-EB	3 mg/kg for 2 weeks	↓Tumor progression, ↓tumor growth, ↓side effects	[56]
Colon	SW620 tumor-bearing nude BALB/c mice	CPT-loaded CMD NPs	10 mg/kg for 30 days	↑Tumor elimination, ↓tumor progression, ↓tumor growth, ↑anticancer efficiency	[92]
Colon	HT29 tumor-bearing nude mice	CPT&Ce6, HBPTK-Ce6@CPT	8 mg/kg for 25 days	↑Tumor elimination, ↓tumor progression, ↓tumor growth, ↓tumor volume	[93]
Colon	Athymic mice bearing HCT116 xenografts	CPT-loaded NPs	1.9 mg/kg for 22 days	↑Tumor elimination, ↓tumor progression, ↓tumor growth	[94]
Colon	HCT116 tumor-bearing mice	CT MLNPs	2.4 ng/kg for 15 days	↑Tumor elimination, ↓tumor progression, ↓tumor growth	[95]
Colon	HCT-116 tumor-bearing nude mice	CPT + ICG NPs	2.5 mg/kg for 55 days	↓Tumor suppression, ↓tumor growth, ↓tumor volume	[96]
Colon	HCT 116 tumor-bearing nude BALB/c mice	CPT-DNA-Gel	1.6 mg/kg for 10–60 days	↓Tumor recurrence, ↑therapeutic efficacy, ↑apoptosis, ↓side effects, ↑tumor inhibition	[97]
Colon	HCT116 tumor-bearing mic	HRC@F127 NPs	5 mg/kg for 40 days	↓Tumor progression, ↓tumor growth, ↑therapeutic efficacy	[98]
Colon	HT-29.Fluc tumor-bearing mice	SNP-CPT and SNP-CPT-Cy5.5 NPs	0.8 mg/kg for 22 days	↑Tumor elimination, ↓tumor progression, ↓tumor growth	[99]
Colon	HCT-8 tumor- bearing BALB/c mice	Nano-CPT	1–2 mg/kg for 5 weeks	↓Tumor suppression, ↓tumor growth, ↓tumor volume	[100]
Colon	Nude mice bearing LoVo tumors	Gd(DTPA-CPT) NPs and Gd(DTPA)/CPT mixture	0.02–0.10 mmol/kg for 0.5 h- 30 days	↑Tumor elimination, ↓tumor growth, ↑anticancer efficiency	[101]
Colon	HT-29 tumor-bearing BALB/c mice	CPT@Dod-ND-SPs	3 mg/kg for 18 days	↓Tumor suppression, ↓tumor growth, ↓tumor volumes	[102]
Colon	C26 tumor-bearing BALB/C mice	PEG-PAMAM-CPT and Apt-PEG-PAMAM-CPT	3 mg/kg for 20 days	↑Tumor elimination, ↓tumor progression, ↓tumor growth	[103]
Colon	BALB/c mice bearing C26 tumors	PEG-PLGA-CPT-NPs or tet-PEG-PLGA- CPT-NPs	10 mg/kg for 20 days	↑Therapeutic index, ↑tumor elimination, ↓tumor progression, ↓tumor growth	[104]
Colon	C26 tumor-bearing Balb/c mice	PEG@MSNR-CPT, PEG@MSNR-CPT/Sur and Apt-PEG@ MSNR-CPT/Sur	3 mg/kg for 35 days	↑Tumor elimination, ↓tumor progression, ↓tumor growth	[105]
Colon	Colon-26 tumor-bearing mice	Fab’-siCD98/CPT-NPs	1.5 mg/kg for 1–50 days	↓Tumorigenesis, ↓tumor growth	[106]
Colon	Colon-26 tumor-bearing mice	PEG-CM-NPs and iRGD-PEG-CM-NPs	3 mg/kg for 10 days	↑Therapeutic efficacy, ↓tumor progression, ↓tumor growth	[107]
Liver	BEL-7402 tumor-bearing nude mice	MGO@CD-CA-HA/CPT	0.1–1 mg/kg for 3 weeks	↑Apoptosis, ↓tumor progression, ↓Tumor growth, ↑therapeutic efficacy, ↑cellular uptake	[108]
Liver	H22 tumor-bearing mice	RGD-prodrug	10 mg/kg for 2 weeks	↑Tumor elimination, ↓tumor growth, ↓tumor volume	[109]
Liver	HepG2 and H22 tumor-bearing mice	CCLM and CCLM without PBA	5 mg/kg for 25 days	↑Tumor elimination, ↓tumor progression, ↓tumor growth	[110]
Liver	balb/c mice bearing Hep1–6 tumor	IR780-LA/CPT-ss-CPT NPs	5.74 μmol/kg for 15 days	↑Synergistic chemo-photothermal therapy, ↑tumor elimination, ↓tumor growth	[111]
Liver	HepG2 tumor-bearing BALB/c nude mice	P(CPT-MAA) prodrug nanogels, P(CPT-MAA) nanogels without SeS	5–10 mg/kg for 21 days	↑Tumor elimination, ↓tumor growth, ↓side effects	[112]
Liver	H22 tumor-bearing mice	JP@EF and JNM@EF	4.0 mg/kg for 40 days	↑Tumor elimination, ↓tumor progression, ↓tumor growth	[113]
Liver	HepG2 tumor-bearing mice	UCNP@mSiO2-NBCCPT/β-CD-PEG, UCNP@mSiO2–NBCCPT @(DHMA)/β-CD-PEG, UCNP@mSiO2-NBCCPT@(DHMA)/β-CD-PEG-LA	15 mg/kg for 21 days	↓Tumor progression, ↓tumor growth, ↓tumor volume, ↑median survival time	[114]
Lung	LLC tumor-bearing mice	CPT-loaded PPBS NPs	5–10 mg/kg for 25 days	↑Tumor elimination, ↓tumor progression, ↓tumor growth	[115]
Lung	A549 tumor-bearing Balb/c mice	NCssG NPs and CssG NWs	4.8 mg/kg for 16 days	↑Tumor elimination, ↓tumor progression, ↓tumor growth, ↓tumor volume	[116]
Lung	A549 tumor-bearing nude mice	CPT-loaded GCMSNs	0.1–0.4 mg for 22 days	↑Tumor elimination, ↓tumor progression, ↓tumor growth, ↑synergistic therapeutic effects	[117]
Lung	BALB/c nude mice bearing A549 tumor	PCPT-V and GOD@PCPT-NR	35 mg/kg for 1–50 days	↓Tumor progression, ↓tumor growth, ↑therapeutic efficacy	[118]
Lung	A549 tumor-bearing mice	ZTC-NMs	3 mg/kg for 1–30 days	↓Tumor progression, ↓tumor growth	[119]
Lung	NCI-H460 tumor-bearing nude mice	Ce6-CPT-UCNPs	2.5 mg/kg for 18 days	↓Tumor recurrence, ↓metastasis, ↓tumor progression, ↓tumor growth	[120]
Lung	NCI-H460 tumor-bearing nude mice	ZnPc/CPT-TPPNPs or ZnPc/CPT-NH2NPs	22.65 mg/kg for 16 days	↓Tumor recurrence, ↑tumor elimination, ↓tumor progression, ↓tumor growth	[121]
Lung	LLC tumor-bearing mice	2OA-CPT/NAs and OA-CPT/NAs	10 mg/kg for 14 days	↑Tumor elimination, ↓tumor progression, ↓tumor growth	[122]
Ovary	SKOV3 tumor-bearing nude mice	CPT-loaded NOBs	0.5 mg/kg for 25 days	↓Tumor suppression, ↓tumor growth, ↓tumor volumes	[123]
Pancreas	BxPC-3 tumor-bearing mice	GSHPTCPT, EGGPTCPT, and PEGPTCPT	10 mg/kg for 12 h–14 days	↓Tumor progression, ↓Tumor growth	[124]
Pancreas	PANC-1 and MIA PaCa-2 xenografts in SCID mice	CPT-loaded nude NPs and CPT-loaded αDR5-NPs	2 mg/kg for 30 days	↑Tumor regressions, ↓tumor growth	[125]
Pancreas	PANC-1 tumor-bearing nude mice	DPPSC + LL, and DPPSC + SL	5 mg/kg for 3–42 days	↓Tumor size, ↓tumor growth, ↑therapeutic efficacy	[126]
Prostate	U14 tumor-bearing mice	CPT-HA and CPT-HA@IR825	1 mg/mL for 12 days	↑Tumor elimination, ↓tumor progression, ↓tumor growth, ↓tumor volume	[127]
Prostate	DU145 tumor-bearing nude mice	P(OEGMA-co-CPT-co-G3-C12) or P(OEGMA-co-CPT)	0.8 mg/kg for 32 days	↓Tumor size, ↑tumor elimination, ↓tumor progression, ↓tumor growth	[128]
Skin (Melanoma)	B16 tumor-bearing nude mice	PEG-SeSe-CPT and PEG-SeSe-CPT/CUR	9.8–10 mg for 21 days	↑Tumor elimination, ↓tumor growth, ↑synergistic therapy effect	[129]

Symbols: ↓, reduction or decrease; *↑,* increase.

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
