# Peer review of "Recent Advances in Improved Anticancer Efficacies of Camptothecin Nano-Formulations: A Systematic Review"

_biomedicines, 2021, doi:10.3390/biomedicines9050480_

Round 1

Reviewer 1 Report

In studies of the 60s, alkaloid camptothecin (isolated from the plant Camptotheca acuminata) demonstrated the ability to bind a complex of DNA and topoisomerase I, preventing the connection of the broken DNA strand and thereby leading to cell death. However, the pronounced toxicity and extremely low solubility suspended further study of the camptothecin. In the 90s, a large number of camptothecin derivatives were synthesized, but only two of them, topotecan and irinotecan, found practical application.

Meanwhile, up to now, intensive research is being carried out in the field of the synthesis of new derivatives of camptothecin, as well as methods for increasing its solubility. Therefore, the review presented by the authors, containing modern literature data, a comprehensive and critical assessment of the antitumor ability of nano-CPT in various cancers as a new and more effective natural compound for drug development is important and relevant.

I believe that the review fully reveals the topic and can be published after minor editing and answering the following questions:

- There is no information in the review whether similar studies on nano-formulations are being carried out with other camptothecin derivatives (topotecan and irinotecan)?

- I believe that the list of cited literature should include references to reviews on topoisomerase inhibitors containing information about camptothecin derivatives (Chem. Rev. 109 (2009) 2894-2902; Nature Rev. 6 (2006) 789-802; Stud. Nat . Prod. Chem. 54 (2017), 21-86; Mol. Cancer Ther. 8 (2009) 1008–1014.), as well as Prof. Yves Pommier and Dr. Mark Cushman’s works in the field of the synthesis of camptothecin derivatives and the study of their properties.

- In vivo and in vitro should be italicized.

- Authors should carefully check the list of cited literature. Some of the references do not have pages (Ref. 140, 141, 131, 130, 127 and many others).

Author Response

The authors of this manuscript express their sincere thanks to the reviewer for the critical assessment of this work. The authors have acted upon the recommendations of the reviewer which have resulted in a significant enhancement in the quality of this manuscript. All modifications incorporated in the manuscript are highlighted in red color font. A “point-by-point” response to each and every comment is outlined below.

General comments:

In studies of the 60s, alkaloid camptothecin (isolated from the plant Camptotheca acuminata) demonstrated the ability to bind a complex of DNA and topoisomerase I, preventing the connection of the broken DNA strand and thereby leading to cell death. However, the pronounced toxicity and extremely low solubility suspended further study of the camptothecin. In the 90s, a large number of camptothecin derivatives were synthesized, but only two of them, topotecan and irinotecan, found practical application.

Meanwhile, up to now, intensive research is being carried out in the field of the synthesis of new derivatives of camptothecin, as well as methods for increasing its solubility. Therefore, the review presented by the authors, containing modern literature data, a comprehensive and critical assessment of the antitumor ability of nano-CPT in various cancers as a new and more effective natural compound for drug development is important and relevant.

I believe that the review fully reveals the topic and can be published after minor editing and answering the following questions:

Response:

We thank the reviewer for their expertise, time, and effort for reviewing our manuscript. We are deeply encouraged by the generous comments about the quality of our work. As described below, we have revised our manuscript based on reviewer’s worthy comments.

Specific comments:

Comment 1:

There is no information in the review whether similar studies on nano-formulations are being carried out with other camptothecin derivatives (topotecan and irinotecan)?

Response:

We greatly appreciate the reviewer’s valuable comments. The information about other nano-formulations of camptothecin derivatives against different cancers already exists in the manuscript (page 2, lines 96-98, reference 34 and page 3, lines 99-102, reference 35).  Additionally, we have added new text regarding irinotecan nano-formulations and cancer to the manuscript (page 19, lines 664 and 665, reference 36 and page 3, lines 102 and 103).

Comment 2:

I believe that the list of cited literature should include references to reviews on topoisomerase inhibitors containing information about camptothecin derivatives (Chem. Rev. 109 (2009) 2894-2902; Nature Rev. 6 (2006) 789-802; Stud. Nat. Prod. Chem. 54 (2017), 21-86; Mol. Cancer Ther. 8 (2009) 1008–1014.), as well as Prof. Yves Pommier and Dr. Mark Cushman’s works in the field of the synthesis of camptothecin derivatives and the study of their properties.

Response:

We believe the reviewer has made an excellent point. Accordingly, additional information about other camptothecin derivatives as topoisomerase inhibitors has been added to the manuscript with citation of the references suggested by the expert reviewer (page 2, lines 55-57 and page 18, lines 613-623, references 14-18).

Comment 3:

In vivo and in vitro should be italicized.

Response:

As recommended by the reviewer, “in vitro” and “in vivo” have been italicized in the manuscript.

Comment 4:

Authors should carefully check the list of cited literature. Some of the references do not have pages (Ref. 140, 141, 131, 130, 127 and many others).

Response:

All the references have been modified and the page numbers have been added to the reference list.

Additionally,

  1. The reference list has been modified and renumbered accordingly. Special attention is given to conform to the order of references and bibliographic style of the journal.
  2. The entire manuscript has been thoroughly checked and edited to ensure uniform style, organization, and quality.

On behalf of my co-authors, I once again express my sincere thanks to the erudite reviewer for the valuable suggestions and constructive input to improve the quality of our manuscript.

Reviewer 2 Report

This manuscript described the results of systematic review for the camptothecin  nano-formulations in cancer therapy. This study identified improved efficacy of camptothecin  nano-formulations over free camptothecin. Authors summarized important studies comprehensively, and provided good evidences for clinical application. This manuscript seems suitable for publication. 

Author Response

Comment:

This manuscript described the results of systematic review for the camptothecin nano-formulations in cancer therapy. This study identified improved efficacy of camptothecin nano-formulations over free camptothecin. Authors summarized important studies comprehensively, and provided good evidences for clinical application. This manuscript seems suitable for publication. 

Response:

We thank the reviewer for their expertise, time, and effort for reviewing our manuscript. We are deeply encouraged by the generous comments about the quality of our work.

Reviewer 3 Report

Camptothecin has been isolated from the tree Camptotheca acuminate small relict tree, which grows in the mountain forests of south-western China. The research team from China in 1985 first identified a marked antitumor action of camptothecin. Camptothecin proved to be extremely effective in the treatment of experimental leukemia in laboratory mice and only 20 years later, it was demonstrated that this alkaloid is a specific topoisomerase I inhibitor, capable of stabilizing the covalent complex of DNA and topoisomerase I.

Extremely poor aqueous solubility, high toxicity and the complexity of the extraction from biological raw materials are major constraints that limit the wide use of camptothecin in medicine as an anticancer drug. Therefore, along with studying the peculiarities of the CPT action, an intensive search for its synthetic analogs and new forms (for example, nano-formulation) with higher hydrophilicity and having the least side effects is carried out.

Considering the above, the review presented by the authors "Recent Advances in Improved Anticancer Efficacies of Camptothecin Nano-Formulations: A Systematic Review" is important, relevant and can be in demand by a wide range of chemists and pharmacologists conducting scientific research in this area and the paper could become suitable for publication after minor revision. Here are some questions and comments for the authors:

- “in situ” and “in vivo” should be italicized

- I believe that the introductory part should contain more references to reviews in the field of synthetic transformations of camptothecin and its activity against human topoisomerases

- the list of cited literature should be unified (wherever possible, doi should be given, in some places incomplete data are given - pages are missing).

Author Response

The authors of this manuscript express their sincere thanks to the reviewer for the critical assessment of this work. The authors have acted upon the recommendations of the reviewer which have resulted in a significant enhancement in the quality of this manuscript. All modifications incorporated in the manuscript are highlighted in red color font. A “point-by-point” response to each and every comment is outlined below.

General comments:

Camptothecin has been isolated from the tree Camptotheca acuminate small relict tree, which grows in the mountain forests of south-western China. The research team from China in 1985 first identified a marked antitumor action of camptothecin. Camptothecin proved to be extremely effective in the treatment of experimental leukemia in laboratory mice and only 20 years later, it was demonstrated that this alkaloid is a specific topoisomerase I inhibitor, capable of stabilizing the covalent complex of DNA and topoisomerase I.

Extremely poor aqueous solubility, high toxicity and the complexity of the extraction from biological raw materials are major constraints that limit the wide use of camptothecin in medicine as an anticancer drug. Therefore, along with studying the peculiarities of the CPT action, an intensive search for its synthetic analogs and new forms (for example, nano-formulation) with higher hydrophilicity and having the least side effects is carried out.

Considering the above, the review presented by the authors "Recent Advances in Improved Anticancer Efficacies of Camptothecin Nano-Formulations: A Systematic Review" is important, relevant and can be in demand by a wide range of chemists and pharmacologists conducting scientific research in this area and the paper could become suitable for publication after minor revision. Here are some questions and comments for the authors:

Response:

We thank the reviewer for their expertise, time, and effort for reviewing our manuscript. We are deeply encouraged by the generous comments about the quality of our work. As presented below, we have revised our manuscript based on the reviewer’s worthy comments.

Specific comments:

Comment 1:

“in situ” and “in vivo” should be italicized

Response:

As recommended by the reviewer, “in situ” and “in vivo” have been italicized in the manuscript.

Comment 2:

I believe that the introductory part should contain more references to reviews in the field of synthetic transformations of camptothecin and its activity against human topoisomerases

Response:

Thank you for this thought-provoking comment. Additional information about synthetic transformations of camptothecin and its activity against human topoisomerases has been added to the manuscript (page 2, lines 53-56).

Comment 3:

The list of cited literature should be unified (wherever possible, doi should be given, in some places incomplete data are given - pages are missing).

Response:

The list of cited literature has been updated and the page numbers and information on doi have been added to the reference list. It is our humble understanding that the journal would reformat the references as needed.

Additionally,

  1. The reference list has been modified and renumbered accordingly. Special attention is given to conform to the order of references and bibliographic style of the journal.
  2. The entire manuscript has been thoroughly checked and edited to ensure uniform style, organization, and quality.

On behalf of my co-authors, I once again express my sincere thanks to the erudite reviewer for the valuable suggestions and constructive input to improve the quality of our manuscript.